# Detecting *Phytophthora cinnamomi* associated with dieback disease on *Carya cathayensis* using loop-mediated isothermal amplification

Xiaoqing Tong[1][☉], Jiayi Wu[1][☉], Li Mei[2], Yongjun Wang[1]*

**1** Department of Forest Protection, College of Forestry and Biotechnology, Zhejiang A&F University, Hangzhou, Zhejiang, China, **2** State Key Laboratory of Subtropical Silviculture, Zhejiang A&F University, Hangzhou, Zhejiang, China

☉ These authors contributed equally to this work.
* wangyj@zafu.edu.cn

**Data Availability Statement:** All relevant data are within the paper and its Supporting information files.

## Abstract

Chinese hickory (*Carya cathayensis* Sarg.) is an economically and ecologically important nut plant in China. Dieback and basal stem necrosis have been observed in the plants since 2016, and its recent spread has significantly affected plant growth and nut production. Therefore, a survey was conducted to evaluate the disease incidence at five sites in Linan County, China. The highest incidence was recorded at the Tuankou site at up to 11.39% in 2019. The oomycete, *Phytophthora cinnamomi*, was isolated from symptomatic plant tissue and plantation soil using baiting and selective media-based detection methods and identified. Artificial infection with the representative *P. cinnamomi* ST402 isolate produced vertically elongated discolorations in the outer xylem and necrotic symptoms in *C. cathayensis* seedlings in a greenhouse trial. Molecular detections based on loop-mediated isothermal amplification (LAMP) specific to *P. cinnamomi* ST402 were conducted. Result indicated that LAMP detection showed a high coherence level with the baiting assays for *P. cinnamomi* detection in the field. This study provides the evidence of existence of high-pathogenic *P. cinnamomi* in the *C. cathayensis* plantation soil in China and the insights into a convenient tool developed for conducting field monitoring of this aggressive pathogen.

## Introduction

Chinese hickory (*Carya cathayensis* Sarg.), belonging to the Juglandaceae family, is an economically important crop in China used for obtaining edible nuts and oil production. Currently, more than 15,000 ha of *C. cathayensis* are cultivated in Zhejiang Province, China, providing income to local farmers and conferring ecological protection in the mountainous areas of eastern China. Traditional cultivation practices have been replaced by monoculture and fertilization to facilitate efficient management and high yield; however, these plant management practices have resulted in several plant diseases, such as *Botryosphaeria* trunk canker [1,2]. In May 2016, a significant outbreak of dieback disease in *C. cathayensis* was recorded in several Linan County orchards that constitute China's main production area. The hickory leaves showed gradual yellowing and wilting, resulting in defoliation and, ultimately, in plant death. Necrotic symptoms were consistently observed on the same side of the basal stem as the

**Funding:** Financial support for this work was provided by grants from Zhejiang Key Research and Development Program of China (2019C0203002) awarded by Wang Y. The funders had no role in study design, data collection and analysis, decision to publish, or preparation of the manuscript.

**Competing interests:** The authors have declared that no competing interests exist.

defoliated leaves. The disease caused large casualty of the nut crop and compromised the incomes of local farmers and the environmental security.

Dieback and basal stem canker diseases in various woody plant hosts caused by diverse oomycete *Phytophthora* species have been recorded in many regions worldwide [3,4]. The genus *Phytophthora* contains over 80 species that exhibit considerable morphological diversity [5]. Among the *Phytophthora* species, *P. cinnamomi* is one of the most aggressive species worldwide, with a broad range of woody plants as the host [6]. The oomycete can grow saprophytically and persist in soil or infected plant materials as oospores or chlamydospores for up to 6 years [3]. The motile zoospores released from the sporangia attach to and invade the roots and basal stems of susceptible hosts, thereby causing necrosis at the infection site. The hyphae form sporangia on the plant surface, resulting in the formation of amplified causal inocula, and the pathogen often causes dieback in young shoots due to interference with root-to-shoot transpiration [7].

The purpose of this study was to determine the existence of the pathogenic *P. cinnamomi* in the soil of diseased plants and to detect the pathogen in the field using loop-mediated isothermal amplification (LAMP); the results could provide insights into an epidemic risk profile and basic reference for the disease management of this pathogenic oomycete.

## Materials and methods

### Field survey

Five commercial *C. cathayensis* orchards in Tuankou, Daoshi, Longgang, Baiguo, and Qinliangfeng in Linan County, Zhejiang Province, China, were selected for the dieback and basal stem canker disease survey (Table 1). Visual disease assessments were performed in July 2018 and July 2019, in which the symptomatic plants were counted based on necrotic cankers present on the basal stem. The disease incidence was presented as the percentage symptomatic plants of the total surveyed plants. The soil pH was directly determined at five randomly selected points in each location using an HI99121 digital Direct Soil Measurement pH Meter (Hanna Instruments, Hanna, Italy).

### *Phytophthora* isolation

The leaf baiting method was used to isolate and detect the presence of zoospore-producing pathogens in soil samples [8]. Briefly, the soil was flooded with a 60-ml aliquot of sterile water in a plastic box and baited with four rhododendron leaves that had been surface-sterilized by wiping with 70% ethanol. The loaded boxes were incubated in a growth chamber at 27°C in the dark for 1 week. Pieces were cut from the leaf lesions and placed in plates containing pimarcin-ampicillin-rifampicin-pentachloronitrobenzene cornmeal agar (PARP) [9]. The hyphal tips grown on the PARP plates were excised and transferred to V8 agar (V8A) medium containing 16 g agar, 3 g $CaCO_3$, 100 ml Campbell's V8 juice, and 900 ml distilled water for

**Table 1. Survey of dieback disease of *Carya cathayensis*.**

| Surveyed location | Longitude and latitude | Altitude (m) | Age of orchard (year) | Mean of soil pH value | Disease incidence (%) | |
|---|---|---|---|---|---|---|
| | | | | | 2018 | 2019 |
| Tuankou | 30˚0'14" N, 119˚7'46' 'E | 350–380 | 30 | 5.23 | 4.43 | 11.39 |
| Daoshi | 30˚17'36" N, 18˚57'44" E | 450–480 | 40 | 5.12 | 3.17 | 5.82 |
| Longgang | 30˚10'27" N, 119˚7'18" E | 390–420 | 30 | 5.46 | 4.23 | 7.04 |
| Baiguo | 30˚4'27" N, 119˚0'15" E | 330–360 | 35 | 5.63 | 0.38 | 0.76 |
| Qinliangfeng | 30˚9'0" N, 118˚57'55" E | 530–550 | 40 | 5.42 | 2.70 | 6.49 |

further investigation. The sporangia and oospores were observed under an optical microscope (Keyence, VHX-6000, Japan).

### *Phytophthora* isolation from symptomatic plant tissue

Segments of necrotic plant tissue were surface-sterilized with 70% ethanol, placed in plates containing the PARP medium, and incubated in a growth chamber at 27˚C in the dark. The hyphal tips grown on the PARP plates were excised and transferred to V8A medium-containing plates.

### DNA extraction and PCR amplification

Genomic DNA (gDNA) extraction from the isolates grown on V8A medium was conducted using the Spin Column Fungal Total DNA Purification Kit (Sangon Biotech, Shanghai), and DNA concentration was determined using the NanoDrop 2000c spectrophotometer (Nano-Drop Technologies, Wilmington, DE, USA). The genomic DNA was used to amplify the ITS region and *COI* gene using ITS4 (TCCTCCGCTTATTGATATGC)/ITS6 (GAAGGTGAAGTCG–TAACAAGG) and COI-Levup (TCAWCWMGATGGCTTTTTTCAAC)/COI-Levlo (CYTCHGGRT GWCCRAAAAACCAAA) primers, respectively [10]. PCR was performed with an initial denaturation at 95˚C for 5 min, followed by 35 cycles of denaturation at 95˚C for 30 s, annealing at 55˚C (ITS4/ITS6 primers) or 52˚C (COI-Levup/COI-Levlo primers) for 30 s, and extension at 72˚C for 1 min, followed by a final extension at 72˚C for 5 min, and a final hold at 4˚C.

### Sequencing and phylogenetic analyses

The amplified PCR products were visualized using 1.2% agarose gel electrophoresis and sequenced (Sangon, Shanghai, China). The data on sequences were queried using the GenBank BLASTn search, and ten representative *Phytophthora* species sequences deposited in the CBS-KNAW culture collection, Netherlands, (http://www.wi.knaw.nl/collections/) were selected for conducting phylogenetic analyses. *Pythium aphanidermatum* was used as the out-group. ITS and *COI* concatenated sequences were generated and aligned using the maximum parsimony method with 1,000 bootstrap replicates in the MAGE 7 software [11].

### Pathogenicity assay

Three-year-old pot-grown Chinese hickory seedlings were used for pathogen inoculation. Briefly, a $2 \times 2$-mm$^2$ wound was inflicted on the seedlings before inoculation. A 5-mm mycelial plug excised from the margin of a 3-day-old V8A-grown culture was placed on the wound and sealed with Parafilm. The mock samples were inoculated with sterile V8A plugs. The inoculated seedlings were incubated in a greenhouse at 25˚C and 85% relative humidity. The Parafilm was removed 1 day post-inoculation (dpi). The seedlings were observed daily, and the necrotic symptoms were recorded. The pathogens were re-isolated from the inoculated tissue and identified based on morphological characteristics and phylogenetic analysis, as described above.

### LAMP primers and reaction

We used the LAMP primers and reaction conditions for *P. cinnamomi* targeting Pcin100006 with minor modifications [12]. The primers used in this study included the forward inner primer (FIP) (5'-CGAAGGACGAGGTGAAGGTGGACGCCCATACATCACATACG), backward inner primer (BIP) (5'-CCGCTACTGCAGACGCCGGCAACTCGGGCAAGATGACTTC), F3 (5'-GTTCTGCGCGATTTGGTTAG), and B3 (5'-GCGGATCTTCCGATCTGGTA). The LAMP

reaction mixtures with pre-reaction addition of hydroxynaphthol blue (HNB) (Sigma-Aldrich, St. Louis, Missouri, USA) were optimized in preliminary tests by modifying the concentration of reagents and incubation duration. The optimal 25-μL reaction included 2.5 μL 10× Thermo-Pol buffer (New England Biolabs, Ipswich, Massachusetts, USA), 2.25 μL each of FIP and BIP (1.8 μM), 1 μL each of F3 and B3 (0.4 μM), 2 μL betaine (0.8 M), 3 μL each of deoxyribonucleotide triphosphate (1.2 mM) and MgSO4 (6 mM), 2 μL HNB (180 mM), 1 μL *Bacillus stearothermophilus* (Bst) DNA polymerase (8 U μl$^{-1}$) (New England Biolabs), 2 μL DNA template, and 1 μL nuclease-free $H_2O$. The reaction mixture was incubated in a water bath set at 64°C for 60 min, and the LAMP results were visually observed and determined by the occurrence of color changes [13]. The reactions that turned blue indicated positive results, suggesting the presence of Pcinn100006 specific to *P. cinnamomi*. Those that turned violet indicated negative results, suggesting the absence of *P. cinnamomi*.

### Field detection of *P. cinnamomi*

An approximately 30-year-old *C. cathayensis* orchard in the Tuankou trial was selected for conduction of soil sampling and *P. cinnamomi* detection. In July 2020, soil samples at 0 to 2 cm depth in a 5-cm ring away from the basal stemof each tree were separately collected and labeled. Each soil sample was thoroughly mixed and separated into two equal subsamples. One subsample was used for DNA extraction, and the other was used for conducting leaf-baiting isolation of *Phytophthora* (as per methods described above). Soil DNA (sDNA) extraction from each subsample was performed directly using the Soil DNA Mini Kit (Tiangen, Beijing, China). The sDNA was diluted to a concentration of 2 ng ml$^{-1}$ before addition to the LAMP reaction, which was performed in triplicate for each subsample. The ITS region and *COI* genes obtained from the isolates were amplified and sequenced. The aligned sequences exhibited over 99% similarity between the candidate strain and ST402, indicating that *P. cinnamomi* was present in the sample.

### Statistical analyses

Tukey's HSD test was used for the separation of mean pathogenicity in *C. cathayensis* seedlings. The statistical analyses were performed using the IBM SPSS Statistics v.19 program (IBM SPSS Inc., USA).

## Results

### Field survey of dieback and basal root rot disease in *C. cathayensis*

The yellowing and wilting of leaves in the surveyed *C. cathayensis* orchards gradually occurred in May 2017, followed by defoliation and dieback of young shoots (Fig 1A). The necrosis extended to the stem and caused the occurrence of basal stem rot and cankers (Fig 1B). In severe cases, the necrotic symptoms occurred at locations up to 1 m above the ground (Fig 1C). The disease was detected in all five survey sites and has been reportedly increasing since 2018, with the highest incidence occurring in the Tuankou trial, where it was 4.43% in July 2018, increasing to 11.39% in July 2019 (Table 1).

### *Phytophthora cinnamomi* isolation and characterization

Fifty-eight isolates were obtained from the symptomatic plant tissues and soil samples. They were morphologically identical to those cultured on the V8A medium plates; the colonies were white, fluffy, and compact, with petaloid or chrysanthemum-like growth patterns (Fig 2A). The V8A isolate mycelia were continuous, and the solitary, caduceus, ovoid to spherical

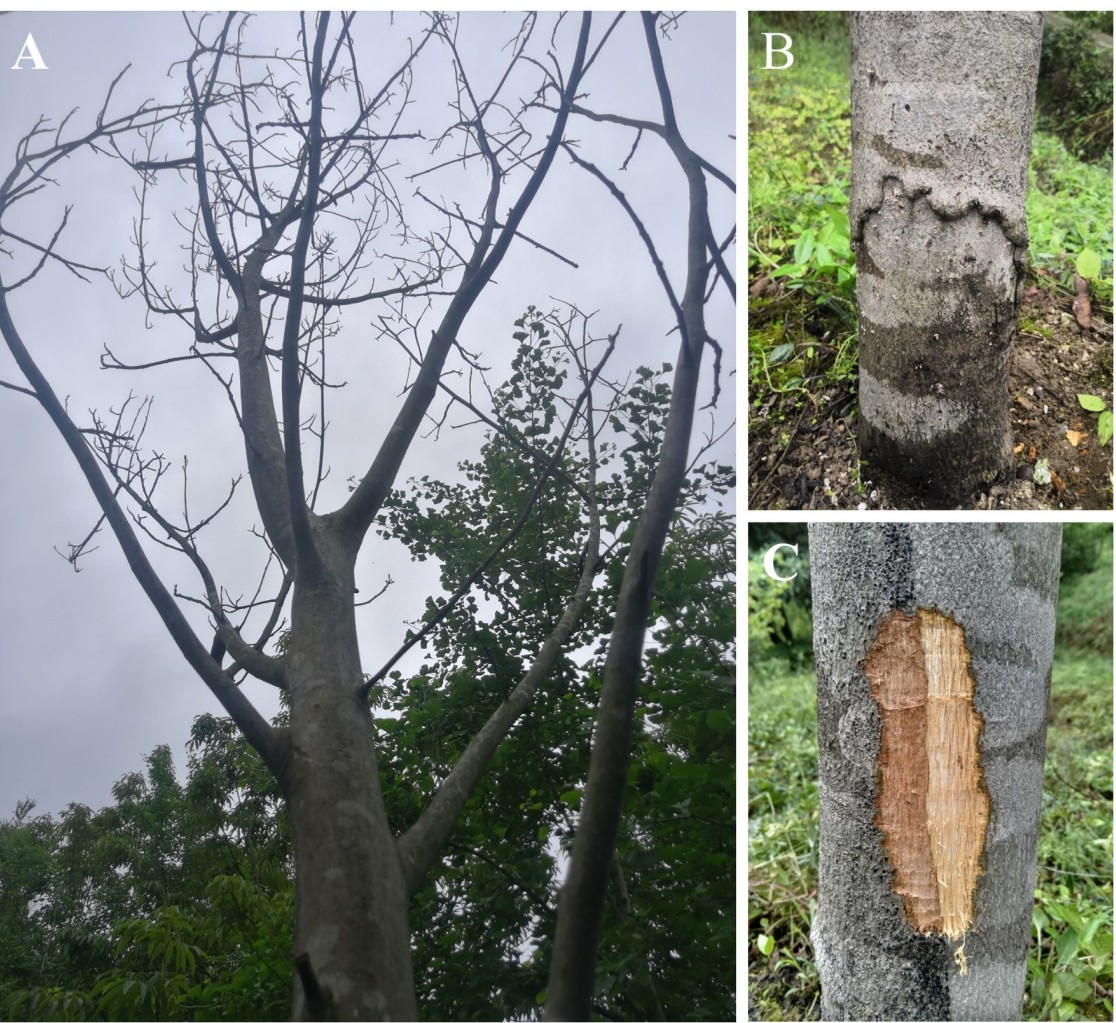

**Fig 1. Dieback and basal stem canker in *Carya cathayensis*.** A: Tree dieback symptoms; B: basal stem necrotic tissue; C: stem-bark extended necrotic zone.

sporangia were produced in a single branch (Fig 2B). The average sporangia size was 21.9 ± 3.8 μm long and 16.2 ± 2.6 μm wide. Globose to slightly subglobose, smooth-walled gametangia were observed at the main hyphae (Fig 2C). The oospores were globose with a mean diameter of 21.6 ± 2.5 μm and were surrounded by a thick wall (Fig 2D). The isolated fungus was preliminarily identified as *P. cinnamomi* based on the above-mentioned morphological characteristics. Three representative isolates were randomly selected for phylogenetic analytic confirmation of the morphological identification. PCR with ITS4/ITS6 and COI-Levup/ COI-Levlo primers resulted in the formation of products of approximately 940 bp and 680 bp for the isolates studied, and all isolates shared identical partial ITS and COI sequences. The ITS and COI sequences of the ST402 isolate were selected and deposited in GenBank with the accession numbers MT675107 and MT683171, respectively. A phylogenetic tree comprising the ST402 isolate and other *Phytophthora* species was constructed based on the concatenated ITS and COI region sequences (Fig 3). The ST402 isolate was distinguished and grouped with *P. cinnamomi*; it was further identified as *P. cinnamomi* by phylogenetic analysis. Moreover, 23 of the 63 isolates obtained from the symptomatic plant

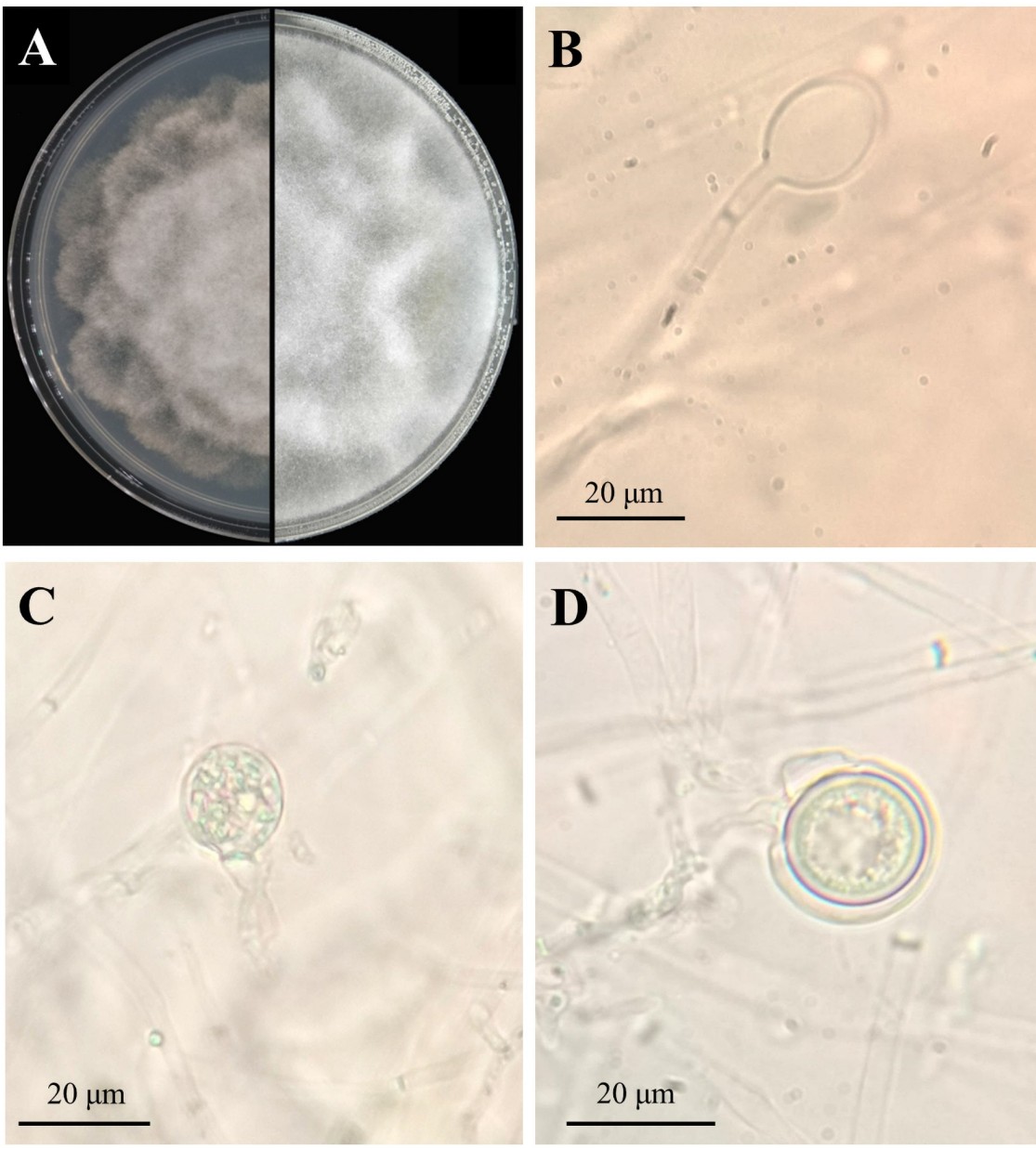

**Fig 2. *Phytophthora cinnamomi* ST402 isolate morphological characteristics.** A: seven-day-old colony on PARP medium; B: sporangia; C: gametangia; D: oospore.

tissues showed identical morphological features and the same ITS and COI sequences as the *P. cinnamomi* ST402 isolate.

### *Phytophthora cinnamomi* exhibited high pathogenicity on *Carya cathayensis* seedlings

The ST402 isolate investigated in the pathogenic tests caused the occurrence of marked necrosis and bark discoloration around the wounded site at 7 dpi in the greenhouse (Fig 4A). The

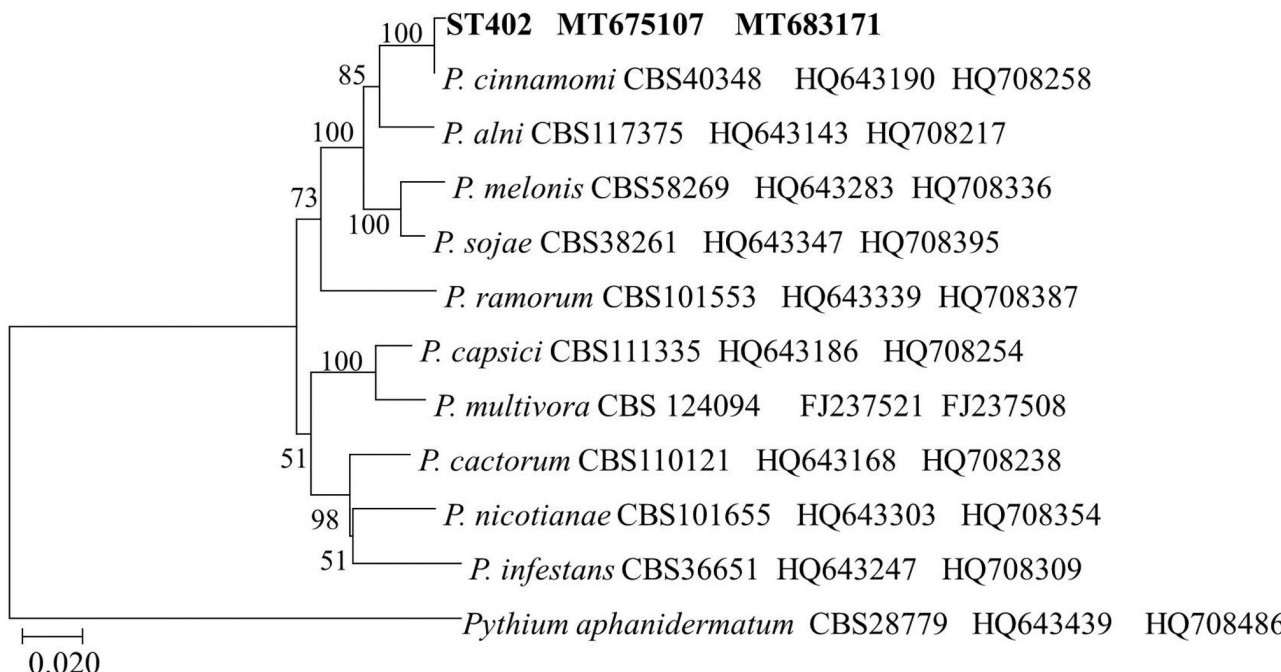

**Fig 3. Phylogenetic analysis of *Phytophthora* species generated using ITS and COI gene sequences.** Phylogenetic tree was rooted to *Pythium aphanidermatum* (CBS 28779) as an outgroup taxon.

3.8 ± 0.4-cm necrosis length caused by *P. cinnamomi* at 7 dpi extended to 5.3 ± 0.6 cm at 14 dpi (Fig 4B). However, no symptoms were observed in the mock inoculations. Re-isolations resulted in the obtainment of colony cultures identical to the morphological and phylogenetic analysis of the *P. cinnamomi* ST402 isolate.

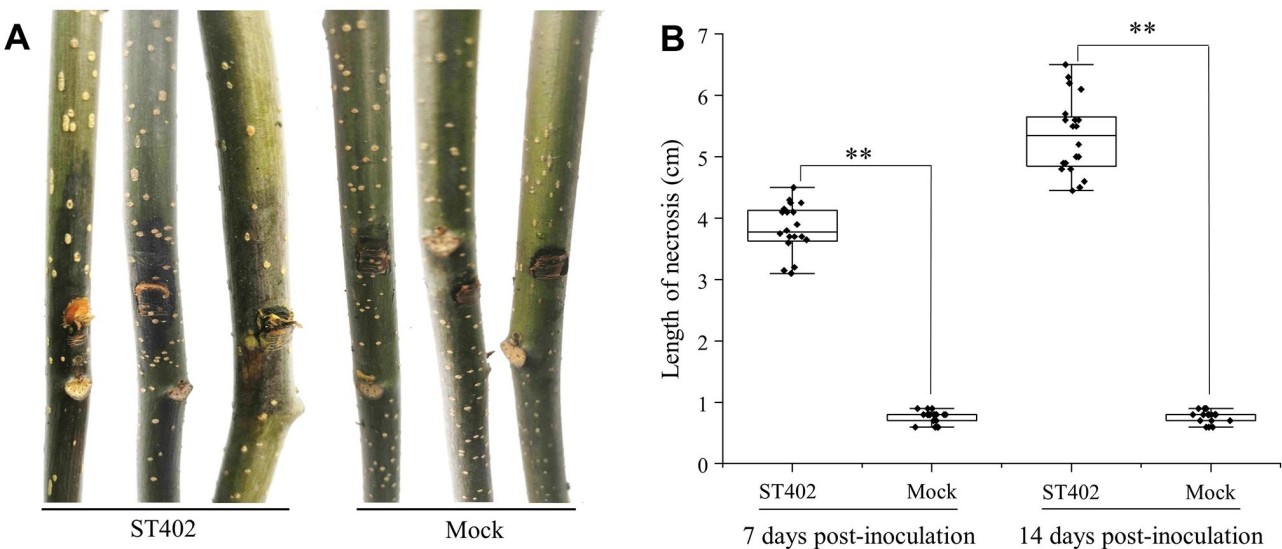

**Fig 4. Pathogenicity of the *Phytophthora cinnamomi* ST402 isolate to *Carya cathayensis* seedlings.** A: *P. cinnamomi* ST402 isolate stem necrosis symptoms 7 days postinoculation; B: Box-plot diagram of *P. cinnamomi* ST402 isolate necrotic length in *C. cathayensis* stems (N = 20) 7 and 14 days postinoculation. Asterisks (*) and (**) indicate significant differences derived from two-sample comparisons at P = 0.05 and P = 0.01, respectively.

### The LAMP detection is specific and sensitive to detect *P. cinnamomi* ST402

The LAMP reaction mixtures containing the ST402 isolate's gDNA exhibited a color change to blue, indicating positive detection (Fig 5A). In contrast, the LAMP reaction mixtures containing gDNAs of other oomycetes or fungi, including *P. cathayensis*, *P. infestans*, *Fusarium oxysporum*, *Pythium ultimum*, and *Verticillium dahlia*, remained purple. Agarose gel visualization of the LAMP reaction mixture showed that only the ST402 isolate exhibited the typical ladder-like pattern. The results of the two independent repeats of each assay were identical.

The LAMP reaction mixtures (25 μL) using 50, 10, 2, 0.4, and 0.08 ng gDNA of the *P. cinnamomi* ST402 isolate yielded positive detection results (Fig 5B). No color change was observed in the reaction mixtures using gDNA at a lower concentration (0.016 ng). In the gel visualization, ladder-like patterns were observed in all reactions; however, the patterns showed less intensity at the lowest concentration (0.016 ng). The results of the two independent repeats of each assay were identical.

### Field detection of *Phytophthora cinnamomi*

Fifteen soil samples from 30 *C. cathayensis* plants showed positive results for baiting assays conducted in July 2019 (Fig 6A). Ten of the thirty plants showed the presence of basal stem

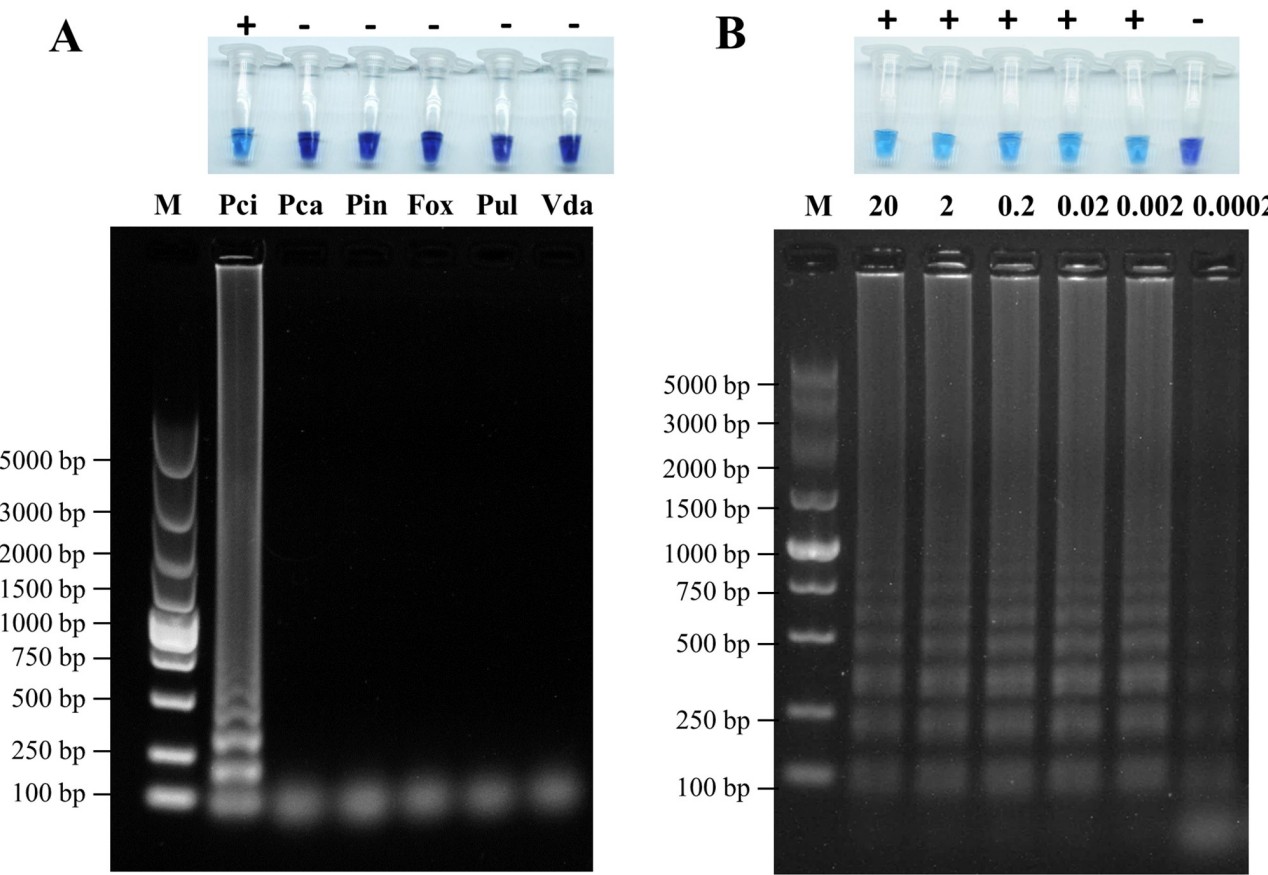

**Fig 5. Specificity (A) and sensitivity (B) of *Pcinn100006* loop-mediated isothermal amplification (LAMP) to *Phytophthora cinnamomi*.** A. LAMP reactions conducted for *P. cinnamomi* specificity and results visualized using hydroxynaphthol blue (upper column) and 1% agarose gel (lower column). M: DNA marker (DL2000, TaKaRa); Pci: *P. cinnamomi* ST402; Pca: *P. cathayensis* CP29; Pin: *P. infestans*; Fox: *Fusarium oxysporum*; Pul: *Pythium ultimum*; Vda: *Verticillium dahliae*. B. LAMP reactions performed for determining *P. cinnamomi* sensitivity and results visualized using hydroxynaphthol blue (upper column) and 1% agarose gel (lower column). *P. cinnamomi* ST402 isolate genomic DNA used as templates (20, 2, 0.2, 0.02, 0.002, and 0.0002 ng per 25 μL reaction mixture). M: DNA marker (DL2000, TaKaRa); +: positive reaction; −: negative reaction.

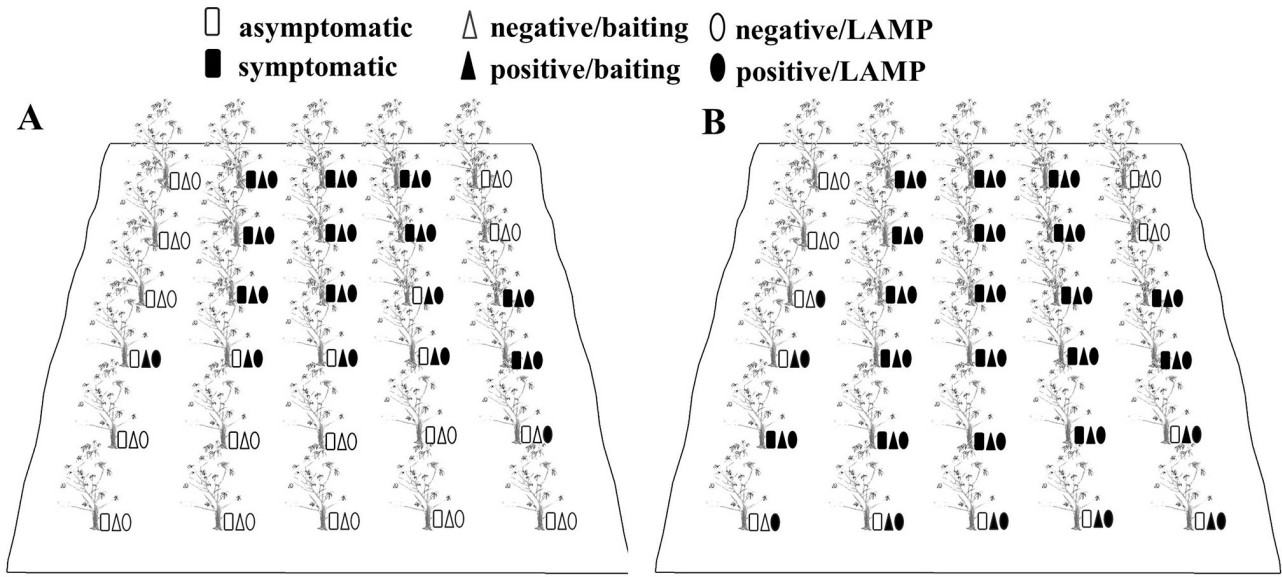

**Fig 6. Detection of *Phytophthora cinnamomi* in *Carya cathayensis* orchard soil samples.** Rhododendron leaf or *P. cinnamomic*-specific LAMP was conducted using DNA samples extracted directly from soil specimens collected from 0 to 5 cm depth around *C. cathayensis* basal stems in a Linan County commercial orchard in July 2019 (A) and July 2020 (B). The hollow and solid squares indicate asymptomatic and necrotic symptoms in *C. cathayensis* plants, respectively. The hollow and solid triangles indicate negative and positive detection by performing baiting assays as per methods described in Materials and methods. Hollow and solid ovals indicate negative and positive detection results by LAMP, respectively.

canker symptoms. All 15 positive baiting-soil samples also showed positive results for LAMP, and one LAMP-positive soil sample was obtained from an asymptomatic plant downstream of the surveyed field. The disease incidence increased in July 2020, with 18 plants exhibiting symptomatic cankers. All soil samples obtained from symptomatic plants showed positive results either in baiting assays or in LAMP-based analyses (Fig 6B). Moreover, eight soil samples obtained from asymptomatic plants showed positive results either in baiting assays or in LAMP-based analyses. These plants were mostly located downstream of the surveyed field.

## Discussion

Several nut tree crops or ornamental trees in the Juglandaceae family are sensitive to *Phytophthora* species. In South Europe, *P. cinnamomi* is frequently isolated from English walnut (*Juglans regia*) trees showing a sudden development of wilt and is considered an aggressive primary pathogen [14,15]. The *P. caryae* species isolated from streams and rivers in the United States are pathogenic to shagbark hickory (*Carya ovata*) [16,17]. In this study, *P. cinnamomi* was isolated from the necrotic tissue and the soil of diseased *C. cathayensis* trees with dieback and basal stem canker symptoms. The ST402 isolate was pathogenic to the host plants in the greenhouse trial. To our knowledge, this is the first report of *P. cinnamomi* associated with dieback and basal stem canker disease on *C. cathayensis*.

*Phytophthora* oomycetes are considered primary pathogens in woody plants worldwide [3,18]. Several species of this genus, such as *P. cinnamomi*, *P. nicotianae*, *P. palmivora*, and *P. ramorum* have very wide host ranges; other species such as *P. alni* and *P. quercina* infect a narrow range of hosts [19]. Moreover, multiple pathogenic species were found in and on the host plants. Six species, namely *P. cinnamomi*, *P. nicotianae*, *P. citricola*, *P. citrophthora*, *P. cryptogea*, and *P. cactorum*, were detected in the same ornamental crop nurseries in the United States [20]. Five species, namely *P. pseudocryptogea*, *P. citrophthora*, *P. nicotianae*, *P. gondwanense*,

and *P. sojae*, were isolated from the rhizosphere of macadamia nut trees in Australia and their ability to cause stem canker was verified [21]. Two *Phytophthora* species, namely *P. cryptogea* and *P. megasperma* caused a decline in the population of wild olive in Spain [22]; furthermore, *P. oleae*, a new pathogen isolated from the rootlets of the same host, showed high pathogenicity, and it was speculated that the three *Phytophthora* species might be active in wild olive roots in different seasons [23]. The new species *P. cathayensis* had been obtained from the active lesions present on the collar of Chinese hickory in a previous study [24]. In this study, *P. cinnamomi* ST402 also exhibited aggressive pathogenicity to the host plants, indicating that the cause of dieback and basal stem canker disease in *C. cathayensis* in China might be due to the *Phytophthora* complex.

The pathogen *P. cinnamomi* is considered as one of the most devastating species and a major threat to natural ecosystems and biodiversity, causing extensive economic losses in agriculture, horticulture, and forestry [25,26]. More than 1,700 plant species, most of which are woody plants, are listed as susceptible to *P. cinnamomi* (http://nt.ars-grin.gov/fungaldatabases/ , April 4, 2021). The wide host range of *P. cinnamomi* has contributed to its rapid spread in several countries worldwide [3]. Moreover, *P. cinnamomi* is known to survive for as long as 6 years in moist soil. The dormant oospores or chlamydospores in contaminated soil and plant materials facilitate its spread through transplanting and irrigation water [27]. It readily inhabits multiple ecological niches that can facilitate its rapid spread and establishment once introduced to a new location [28]. Our results provide evidence that *P. cinnamomi* exists and spreads in *C. cathayensis* plantation soil. However, its origin remains unknown, and its risk in other plants and the environment should be evaluated.

Accurate pathogen detection has critical implications for disease management, disease-free certification, and quarantine. Particularly, mapping of the incidence and distribution of *P. cinnamomi* in plantation soils may offer valuable information for limiting its spread [29]. Currently, *Phytophthora* can be detected using a variety of techniques, including direct isolation on *Phytophthora*-selective media [9], baiting, and isolation onto selective media [30], immunodetection assay [31], and molecular detection based on specific DNA sequences [8,32]. Touchdown nested multiplex PCR targeting ITS sequences was used to detect the presence of *P. cinnamomi* in chestnut grove soils [33]. Conventional methods, including direct isolation from the diseased material, or indirect baiting using known host plants, are labor-intensive and time-consuming [34]. In contrast, molecular genetic methods are relatively more sensitive and time-effective, thereby allowing for a higher throughput. In this study, the combination of rhododendron leaf baiting and LAMP assay was performed in the field to detect the presence of *P. cinnamomi* in plantation soil samples from *C. cathayensis*. Our results revealed the route of transmission of the pathogen in the field, and this revelation could contribute to disease management and the early detection of the pathogen in plantation soils of asymptomatic trees, which might facilitate timely disease prevention.

LAMP is based on the principle of autocycling strand displacement DNA synthesis performed by *Bst* DNA polymerase to detect a specific DNA sequence [35]. This technique is performed using four to six primers that recognize six to eight regions of the target DNA with an extremely high specificity. The technique can be performed under isothermal conditions ranging from 60 °C to 65 °C; it produces a substantial quantity of DNA, thus facilitating detection by DNA dyes, such as SYBR Green I [36]. Importantly, the PCR thermocycler equipment is not required because the temperature is not cycled during the amplification process. LAMP technology has been successfully applied to detect plant pathogenic fungi [37], oomycetes [38], bacteria [39], and nematodes [40]. Recently, the LAMP assay targeting *Pcinn100006* showed highly specific results for *P. cinnamomi* [12]. The *Pcinn100006* loci were also located in the *P. cinnamomi* ST402 genome (Wang et al, unpublished data). The results of LAMP assays

conducted in the present study verified its high specificity and sensitivity in *P. cinnamomi* detection. Moreover, in this study, the LAMP assay was practically utilized to detect *P. cinnamomi* in *C. cathayensis* plantation soil.

## Conclusion

The present study revealed that *P. cinnamomi* isolated from necrotic tissue and diseased *C. cathayensis* plantation soil exhibited high virulence. To our knowledge, *P. cinnamomi* associated with dieback and basal stem canker of *C. cathayensis* was determined for the first time. The molecular detection of *P. cinnamomi* based on LAMP provided the route of transmission of the aggressive pathogen in the field, and may contribute to an early diagnosis of the disease and may help prevent its spread.

## Supporting information

**S1 Table. GenBank accession numbers of taxa used in phylogenetic analyses.**
(DOCX)

**S1 Raw images.**
(PDF)

## Acknowledgments

We thank Dr. Carmen Morales-Rodríguez (DIBAF, University of Tuscia, Italy) and Dr. Andrea Vinnini (DIBAF, University of Tuscia, Italy) for their technical assistances and critical review.

## Author Contributions

**Conceptualization:** Yongjun Wang.

**Data curation:** Xiaoqing Tong, Li Mei.

**Formal analysis:** Jiayi Wu, Li Mei.

**Funding acquisition:** Yongjun Wang.

**Investigation:** Xiaoqing Tong, Jiayi Wu.

**Methodology:** Xiaoqing Tong, Jiayi Wu, Li Mei, Yongjun Wang.

**Project administration:** Yongjun Wang.

**Resources:** Yongjun Wang.

**Software:** Xiaoqing Tong, Jiayi Wu, Li Mei.

**Validation:** Xiaoqing Tong, Jiayi Wu.

**Visualization:** Xiaoqing Tong, Jiayi Wu.

**Writing – original draft:** Li Mei, Yongjun Wang.

**Writing – review & editing:** Yongjun Wang.

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
