## [Decision Letter · Decision Letter 0]

21 Oct 2021

Detecting Phytophthora cinnamomi associated with dieback disease on Carya cathayensis using loop-mediated isothermal amplification

PONE-D-21-28959

Dear Dr. Wang,

We’re pleased to inform you that your manuscript has been judged scientifically suitable for publication and will be formally accepted for publication once it meets all outstanding technical requirements.

Kind regards,

Sagheer Atta, Ph.D

Academic Editor

PLOS ONE

1. Thank you for stating the following financial disclosure: 

 [The funders had no role in study design, data collection and analysis, decision to publish, or preparation of the manuscript.]

a) Please provide an amended Funding Statement that declares *all* the funding or sources of support received during this specific study (whether external or internal to your organization) as detailed online in our guide for authors at http://journals.plos.org/plosone/s/submit-now.  

b) Please state what role the funders took in the study.  If any authors received a salary from any of your funders, please state which authors and which funder. If the funders had no role, please state: "The funders had no role in study design, data collection and analysis, decision to publish, or preparation of the manuscript." 

Please send your amended statements by return email; we will change the online submission form on your behalf. 

Reviewers' comments:

Reviewer's Responses to Questions

**Comments to the Author**

1. Is the manuscript technically sound, and do the data support the conclusions?

Reviewer #1: Yes

Reviewer #2: Yes

2. Has the statistical analysis been performed appropriately and rigorously? 

Reviewer #1: Yes

Reviewer #2: Yes

3. Have the authors made all data underlying the findings in their manuscript fully available?

Reviewer #1: Yes

Reviewer #2: Yes

4. Is the manuscript presented in an intelligible fashion and written in standard English?

Reviewer #1: Yes

Reviewer #2: Yes

5. Review Comments to the Author

Reviewer #1: The research topic/work taken is based on the importance, right now the pathogen is wide host range, moreover it has to be early diagnosed. Experiments were planned very well. The conclusion match with the output of the research work.

Statistical analysis does not need for the present research work. Data required are included in the current research work. Manuscript presented in an intelligible fashion and written in standard English.

Reviewer #2: the manuscript was presented in an intelligible fashion and written in standard English

the statistical analysis has been performed appropriately and rigorously

the manuscript technically is sound, and do the data support the conclusions

the manuscript can be accepted in plos one.

6. PLOS authors have the option to publish the peer review history of their article (what does this mean?). If published, this will include your full peer review and any attached files.

Reviewer #1: **Yes: **A Ramanathan

Reviewer #2: **Yes: **Dr. Sakineh Abbasi

---

## [Editor Report · Acceptance letter]

5 Nov 2021

PONE-D-21-28959 

Detecting *Phytophthora cinnamomi* associated with dieback disease on *Carya cathayensis* using loop-mediated isothermal amplification 

Dear Dr. Wang:

I'm pleased to inform you that your manuscript has been deemed suitable for publication in PLOS ONE. Congratulations! Your manuscript is now with our production department. 

Kind regards, 

on behalf of

Dr. Sagheer Atta 

Academic Editor

PLOS ONE